# Crosslinking by Click Chemistry of Hyaluronan Graft Copolymers Involving Resorcinol-Based Cinnamate Derivatives Leading to Gel-like Materials

**DOI:** 10.3390/gels10110751

**Published:** 2024-11-19

**Authors:** Mario Saletti, Simone Pepi, Marco Paolino, Jacopo Venditti, Germano Giuliani, Claudia Bonechi, Gemma Leone, Agnese Magnani, Claudio Rossi, Andrea Cappelli

**Affiliations:** Dipartimento di Biotecnologie, Chimica e Farmacia, Università degli Studi di Siena, Via Aldo Moro 2, 53100 Siena, Italy; mario.saletti2@unisi.it (M.S.); simone.pepi@unisi.it (S.P.); paolino3@unisi.it (M.P.); jacopo.venditti@student.unisi.it (J.V.); giuliani5@unisi.it (G.G.); claudia.bonechi@unisi.it (C.B.); gemma.leone@unisi.it (G.L.); agnese.magnani@unisi.it (A.M.); claudio.rossi@unisi.it (C.R.)

**Keywords:** hydrogel, gel-like materials, hyaluronic acid, ferulic acid, click chemistry, crosslinking

## Abstract

The well-known “click chemistry” reaction copper(I)-catalyzed azide-alkyne 1,3-dipolar cycloaddition (CuAAC) was used to transform under very mild conditions hyaluronan-based graft copolymers **HA(270)-FA-Pg** into the crosslinked derivatives **HA(270)-FA-TEGERA-CL** and **HA(270)-FA-HEGERA-CL**. In particular, medium molecular weight (i.e., 270 kDa) hyaluronic acid (HA) grafted at various extents (i.e., 10, 20, and 40%) with fluorogenic ferulic acid (FA) residue bonding propargyl groups were used in the CuAAC reaction with novel azido-terminated crosslinking agents **T**ri(**E**thylene **G**lycol) **E**thyl **R**esorcinol **A**crylate (**TEGERA**) and **H**exa(**E**thylene **G**lycol) **E**thyl **R**esorcinol **A**crylate (**HEGERA**). The resulting **HA(270)-FA-TEGERA-CL** and **HA(270)-FA-HEGERA-CL** materials were characterized from the point of view of their structure by performing NMR studies. Moreover, the swelling behavior and rheological features were assessed employing TGA and DSC analysis to evaluate the potential gel-like properties of the resulting crosslinked materials. Despite the 3D crosslinked structure, **HA(270)-FA-TEGERA-CL** and **HA(270)-FA-HEGERA-CL** frameworks showed adequate swelling performance, the required shear thinning behavior, and coefficient of friction values close to those of the main commercial HA solutions used as viscosupplements (i.e., 0.20 at 10 mm/s). Furthermore, the presence of a crosslinked structure guaranteed a longer residence time. Indeed, **HA(270)-FA-TEGERA-CL-40** and **HA(270)-FA-HEGERA-CL-40** after 48 h showed a four times greater enzymatic resistance than the commercial viscosupplements. Based on the promising obtained results, the crosslinked materials are proposed for their potential applicability as novel viscosupplements.

## 1. Introduction

Hyaluronic acid (HA, hyaluronan) is a glycosaminoglycan derivative [1] obtained through a condensing reaction that involves glucuronic acid and *N*-acetylglucosamine linked by β[1-4] and β[1-3] alternate glycosidic bonds [2]. HA is involved in the formation of the pericellular coat, which was suggested to affect the early stages of cell adhesion [3] by interacting with the CD44 receptor [4], a multifunctional transmembrane glycoprotein spread over in almost all human cell types [5]. The activation of HA-CD44 signaling pathways controls several cell functions, including cell proliferation [6], migration [7], aggregation [8], adhesion to extracellular matrix (ECM) components [9,10], and angiogenesis [11,12]. Moreover, HA plays an important role in different physiological functions in human organisms such as wound healing [13,14,15], tissue repair [16,17,18,19], and damping of excessive loads on the joints [20,21]. Owing to its ability to bond a large amount of water [22,23], HA controls tissue hydratation and constitutes a lubricant viscosupplement for healing osteoarticular diseases [24] and eye diseases [25,26,27]. HA is still being studied to reveal its biosynthetic pathways and improve its biotechnological production [28,29] to obtain innovative hyaluronan derivatives with enriched properties in the development of drug delivery systems (DDSs) [20,30]. Particularly, HA (crosslinked or non-crosslinked) derivatives have been widely used in developing biocompatible hydrogels for biomedical and pharmaceutical applications [31,32,33,34,35,36,37,38,39,40].

Ferulic acid (FA, 4-hydroxy-3-methoxycinnamic acid) is a cinnamic acid derivative endowed with several biological activities [41] including a powerful antioxidant property [42,43,44,45] because it is a free radical scavenger [46,47] and an inhibitor of enzymes that catalyze free radical generation. Furthermore, FA possesses potential health benefits [48] and several physiological functions, ranging from anti-inflammatory activity to antidiabetic and anti-hepatotoxic effects [49,50,51,52]. It is also extensively used thanks to its sunscreen feature properties [53]. FA is distributed in the plant cell wall, where it is linked by means of an ester bond to the primary alcoholic function of arabinose side chains, and is involved in crosslinking polysaccharides [54] and proteins during cell wall synthesis [55,56,57,58,59,60].

Today, one of the most emerging strategies in the biomedical sciences and in innovative materials development is represented by click chemistry due to its efficiency, selectivity, and tolerance towards several solvents and functional groups [61]. Especially, Cu(I)-catalyzed azide–alkyne cycloaddition (CuAAC) is the most widely used, finding its application in various fields of chemistry like drug delivery [62,63,64,65,66], polymer chemistry [67], and bioconjugation [68,69]. Overall, this wide use is due to the simplicity of the inherent preparation methods of click chemistry, which allows researchers to deal with more complex systems. Indeed, this synthetic technique has been exploited for several applications [70,71,72], especially for preparing hydrogels [73,74,75,76,77,78,79,80,81,82].

Some years ago, we envisioned HA macromolecules could be grafted with FA fluorophore residues bearing propargyl groups to obtain **HA-FA-Pg** hyaluronan derivatives [83]. These graft copolymers were then employed in the development of a tri-component polybenzofulvene cylindrical brush by means of a convergent approach exploiting a copper(I)-catalyzed azide–alkyne 1,3-dipolar cycloaddition (CuAAC) [84,85].

Encouraged by the obtained results, the coating procedure by using **HA-FA-Pg** graft copolymers was gradually improved to become a versatile technology platform potentially useful in the coating of the surfaces of different nanostructures. These include small unilamellar vesicles (SUVs) [86], self-assembling micelles [87], magnetic nanoparticles [88], and poly(propylene imine) (PPI) dendrimers [89]. Subsequently, a crosslinking strategy involving **HA-FA-Pg** graft copolymers was developed in our lab by using the well-known click chemistry reaction (click-crosslinking) (Appendix A) [90,91].

In the first approach of this strategy, the terminal azide groups of a hexa(ethylene glycol) derivative were involved in the CuAAC coupling with the clickable alkyne moieties of **HA-FA-Pg** graft copolymers, which showed a distinctive molar mass and grafting degree values. In this way, we obtained crosslinked HA derivatives (i.e., HA-FA-HEG-CL) that represent innovative hydrogels [90]. On the other hand, the results of the first click-crosslinking approach motivated the investigation of more complex crosslinking agents showing a greater length and/or bearing aromatic spacers such as the catechol moieties of caffeic acid derivatives as those present in HA(270)-FA-TEGEC-CL and HA(270)-FA-HEGEC-CL materials [91].

In the current work, with the aim of enlarging the application of **HA-FA-Pg** graft copolymers to the development of novel gel-like materials, the clickable crosslinking agents **1a** [azido-terminated **T**ri(**E**thylene **G**lycol) **E**thyl **R**esorcinol **A**crylate, **TEGERA**] and **1b** [azido-terminated **H**exa(**E**thylene **G**lycol) **E**thyl **R**esorcinol **A**crylate, **HEGERA**] were used in the CuAAC reaction with medium-molar-mass-value (i.e., Mw = 270 kDa) graft copolymers **HA(270)-FA-Pg** characterized by different grafting degree values (i.e., 10, 20, and 40%) to afford the respective crosslinked HA derivatives [i.e., **HA(270)-FA-TEGERA-CL** and **HA(270)-FA-HEGERA-CL**, Figure 1], revealing distinctive crosslinking densities.

The physicochemical and rheological properties of the resulting gel-like materials were then characterized, and the derivatives showing the most promising features were studied for their potential applicability as viscosupplements.

## 2. Results and Discussion

### 2.1. Preparation of HA(270)-FA-TEGERA-CL and HA(270)-FA-HEGERA-CL Crosslinked Materials

The first step of the synthetic work was represented by the development of a convergent procedure (Figure 1) for the preparation of the appropriate clickable crosslinking agents **1a** (**TEGERA**) and **1b** (**HEGERA**).

Thus, the appropriate oligo(ethylene glycol) derivatives **3a**,**b** were prepared in three steps [91] from the commercially available tri(ethylene glycol) **2a** or hexa(ethylene glycol) **2b**, and were exploited in the alkylation of the resorcinol moieties of ethyl cinnamate **6** by means of a base, such as cesium carbonate, and sodium iodide as the catalyst to obtain compounds **1a**,**b** in acceptable yields. Derivative **5** was easily synthesized from commercially available *trans*-3,5-dimethoxy cinnamic acid **4**, which was involved in an esterification reaction with phosphorus(V) oxychloride in ethanol to afford the corresponding ethyl ester **5**, followed by demethylation with boron tribromide to liberate the resorcinol scaffold of compound **6**. The clickable groups resulting in crosslinking agents **1a**,**b** were then used in the CuAAC reaction [62,92] with **HA(270)-FA-Pg** graft copolymers showing different grafting degree values (i.e., 10%, 20%, and 40%) to obtain crosslinked HA derivatives (i.e., **HA(270)-FA-TEGERA-CL** and **HA(270)-FA-HEGERA-CL**) showing different crosslinking densities (Figure 2).

Different **HA(270)-FA-Pg** samples (with grafting degree values of 10, 20, and 40%) were used in the click-crosslinking reaction to obtain the respective **HA(270)-FA-TEGERA-CL** and **HA(270)-FA-HEGERA-CL** materials showing different crosslinking groups and crosslinking densities that could affect the flexibility of the HA(270) backbone in the crosslinked materials. With the aim of inducing the exhaustive reaction of the propargyl groups present on the ferulate residues grafting the HA backbones of **HA(270)-FA-Pg** graft copolymers, the amounts of azido derivatives **1a**,**b** were planned. Moreover, the catalytic species, i.e., copper(I), was obtained from CuSO_4_ by reduction with sodium ascorbate to accomplish the CuAAC reaction under very mild conditions.

### 2.2. Structure of HA(270)-FA-TEGERA-CL and HA(270)-FA-HEGERA-CL Materials

^1^H NMR spectroscopy studies were performed on the newly synthesized crosslinked materials **HA(270)-FA-TEGERA-CL** and **HA(270)-FA-HEGERA-CL** using D_2_O as the solvent to gain information about their structures. As previously reported [89,90,91], the occurrence of the click chemistry crosslinking reaction was ascertained by comparing the ^1^H NMR spectra of the crosslinked materials to that of their corresponding synthetic precursors. As an example, Figure 2 shows the comparison of the ^1^H NMR spectrum of **HA(270)-FA-TEGERA-CL-20** with the corresponding spectrum of the starting graft copolymer **HA(270)-FA-Pg-20**.

As previously observed in related crosslinked materials [90,91], the comparison shown in Figure 2, referring to the spectrum of **HA(270)-FA-TEGERA-CL-20** material with respect to its corresponding synthetic precursor **HA(270)-FA-Pg-20**, suggests the presence of a remarkable line broadening. This result could be easily explained in terms of lower mobility in the crosslinked macromolecules of the material with respect to the one of the starting **HA(270)-FA-Pg-20** graft copolymer. Owing to this line broadening, the peaks in the aromatic region of the spectrum of the crosslinked material were very broad and of difficult assignment. However, the presence of a very broad singlet at around 8 ppm assigned to the triazole proton could be detected. This result confirmed the occurrence of the CuAAC reaction [89,90,91].

Slightly better results were obtained with crosslinked material **HA(270)-FA-HEGERA-CL-20** (Figure 3) obtained by using **1b** as the crosslinking agent.

Probably, the greater length of the crosslinking agent **1b** allowed a greater local mobility in the HA chains of the crosslinked material, producing a slightly better line shape in the spectrum of crosslinked material **HA(270)-FA-HEGERA-CL-20**. Thus, the singlet at around 8 ppm assigned to the triazole proton could be easily detected, substantiating the occurrence of the CuAAC reaction. Similar results were obtained with crosslinked materials **HA(270)-FA-TEGERA-CL-10** and **HA(270)-FA-HEGERA-CL-10** (Appendix A), whereas the quality of ^1^H NMR spectra obtained with highly crosslinked materials **HA(270)-FA-TEGERA-CL-40** and **HA(270)-FA-HEGERA-CL-40** was low, probably because of solubility issues.

### 2.3. Physical Characterization of HA(270)-FA-TEGERA-CL and HA(270)-FA-HEGERA-CL Materials

#### 2.3.1. Swelling Behavior

The ability of the developed gel-like materials to swell without dissolving is the first easy method to confirm the crosslinking procedure. The **HA(270)-FA-TEGERA-CL** and **HA(270)-FA-HEGERA-CL** gel-like materials reached swelling equilibrium within 48 h (Appendix A), whereas the previously published HA(270)-FA-HEG-CL series reached equilibrium within 24 h [90]. Significant differences were observed during the first 8 h of swelling. Specifically, the **HA(270)-FA-TEGERA-CL** series exhibited faster swelling kinetics, absorbing around 88% of the total water within the first hour, while the **HA(270)-FA-HEGERA-CL** series showed slower swelling kinetics, absorbing only around 76% of the total water during the same period. This different behavior suggests a variation in compactness among the three series, with the HA(270)-FA-HEG-CL series showing the lowest stiffness and the **HA(270)-FA-HEGERA-CL** series exhibiting the highest compactness. The water absorption process, as explained by Li et al. [93] can be seen as a three-step mechanism. The first step consists in the formation of a shell of water molecules strictly bound to the hydrophilic groups of the gel-like materials (not freezing water: _nf_W). The presence of this shell permits the formation of a second shell, bound to the first one (freezing bound water: _fb_W). This second shell permits the matrix to open itself and to absorb a great amount of water (freezing free water: _ff_W). The three kinds of water give the total water. In Table 1, all the obtained values are summarized.

Considering the swelling behavior in terms of total water uptake, it can be observed that the most significant effect is due to the polymer crosslinking density rather than the type of crosslinking arm. In each series, the frameworks prepared using the material with the lowest crosslinking density exhibited the highest swelling ratio (SR%). However, some differences can also be noted across the different series, specifically a decrease in total water uptake when shifting from the HA(270)-FA-HEG-CL series to the **HA(270)-FA-TEGERA-CL** and **HA(270)-FA-HEGERA-CL** series. This trend suggests an increase in crosslinking efficiency with longer crosslinking arms. In terms of water types, only the **HA(270)-FA-HEGERA-CL** series samples obtained from high-grafting-degree polymers [i.e., **HA(270)-FA-Pg-20** and **HA(270)-FA-Pg-40**] showed an opposite behavior, being characterized by a very high amount of non-freezing water. This peculiar behavior could be related to the shape of the framework since both **HA(270)-FA-HEGERA-CL-20** and **HA(270)-FA-HEGERA-CL-40** appeared as swollen microparticles.

The mesoscale bulk porosity, which can deeply affect the capability of a material to swell, was also calculated. Despite the crosslinking arms used to obtain gel-like materials starting from the highest functionalized graft copolymer (i.e., **HA(270)-FA-Pg-40**), very similar mesoporosity values were obtained. Indeed, HA(270)-FA-HEG-CL-40 showed a pore radius of about 7 nm against 6 nm found for both **HA(270)-FA-TEGERA-CL-40** and **HA(270)-FA-HEGERA-CL-40**. Contrarily, higher variability was found among frameworks obtained starting from both graft copolymers **HA(270)-FA-Pg-10** and **HA(270)-FA-Pg-20**. Indeed, comparing frameworks obtained starting from **HA(270)-FA-Pg-10,** we found for both HA(270)-FA-HEG-CL-10 and **HA(270)-FA-HEGERA-CL-10** a high pore radius (38 nm and 32 nm, respectively), whereas **HA(270)-FA-TEGERA-CL-10** showed a very reduced pore radius similar to that obtained for the 40 series (i.e., 6 nm). Among the **HA(270)-FA-Pg-20** series, only HA(270)-FA-HEG-CL-20 showed a high pore radius (i.e., 36 nm) whereas **HA(270)-FA-HEGERA-CL-20** and **HA(270)-FA-TEGERA-CL-20** showed a pore radius of 12 nm and 9 nm, respectively.

#### 2.3.2. Thermal Stability

The thermographs of the dry starting graft **HA(270)-FA-Pg** copolymers and their crosslinked forms were recorded and compared. Very interestingly, moving from the graft copolymers to their different crosslinked forms [i.e., HA(270)-FA-HEG-CL, **HA(270)-FA-TEGERA-CL**, and **HA(270)-FA-HEGERA-CL** series], a decrease in the temperature at which the highest weight loss is recorded (242 ± 3 °C) can be observed. No significant difference was found between the crosslinked forms, which ranged around 232 ± 3 °C. From a quantitative point of view, the crosslinking process deeply affected the thermal stability with a significant decrease (i.e., ~10%) of total loss in the 30–850 °C range (Appendix A). No significant effects of the crosslinking arms can be observed. The **HA(270)-FA-TEGERA-CL** and **HA(270)-FA-HEGERA-CL** frameworks were considerably more thermally stable in comparison with the HA(270)-FA-HEG-CL frameworks, thus confirming that the use of a longer crosslinking arm positively affected the crosslinking degree.

#### 2.3.3. Rheological Analysis

The mechanical properties of crosslinked gel-like materials were studied by measuring the storage (G′) and the loss (G″) moduli as a function of frequency. The frequency sweep test evidenced a “gel-like” behavior. All the frameworks showed G′ higher than G″ across the entire frequency range analyzed. Furthermore, with G′ being higher than G″ by about one order of magnitude, all the obtained materials can be defined as strong gels (Figure 4).

The highest effect was found shifting from HA(270)-FA-HEG-CL series to **HA(270)-FA-TEGERA-CL** and **HA(270)-FA-HEGERA-CL** frameworks that showed doubled complex modulus G* values. Comparing **HA(270)-FA-TEGERA-CL** and **HA(270)-FA-HEGERA-CL** series, no significant difference was found, whereas it was found in terms of the starting graft copolymer. Indeed, both **HA(270)-FA-TEGERA-CL-10** and **HA(270)-FA-HEGERA-CL-10** had significantly lower elastic moduli than frameworks obtained by **HA(270)-FA-Pg-20** and **HA(270)-FA-Pg-40** (Figure 5).

Based on the stronger mechanical properties of the **HA(270)-FA-TEGERA-CL** and **HA(270)-FA-HEGERA-CL** series in comparison with HA(270)-FA-HEG-CL series, all the obtained gel-like materials were also tested in terms of viscosity, compression resistance, and coefficient of friction to foresee their application as a viscosupplement.

As reported by Rebenda et al. [94] in their work on the rheological and frictional characterization of various commercial viscosupplements, an ideal viscosupplement, designed to act as a joint lubricant and “shock absorber”, must exhibit shear-thinning behavior. All our samples exhibited perfect shear-thinning behavior, with a marked decrease in viscosity as the shear rate increased (Appendix A). Furthermore, the **HA(270)-FA-TEGERA-CL-10** at a shear rate of 1/s exhibited viscosity values similar to those of Synvisc™, one of the most commonly used commercial viscosupplements.

Fakhari et al. measured the rheological moduli of viscosupplements and emphasized that viscosupplements should exhibit an elastic modulus greater than the viscous modulus (G′ > G″). Additionally, the absolute values of these moduli must exceed those of human synovial fluid (G′: 23 Pa, G″: 7 Pa) to increase the residence time after injection [95]. All the prepared frameworks demonstrated high resistance not only to shear stress but also to compression loads, as no crossover point was observed over the analyzed frequency range (Figure 6). In fact, all frameworks maintained a strong elastic component at frequency values typical of normal daily activities (walking: 0.5 Hz, running: 2.5–3 Hz) [95].

#### 2.3.4. Tribological Properties

In their study on the frictional and rheological properties of several commercial viscosupplements, Bonnevie et al. [96] found that friction measurements were more predictive of clinical performance than viscoelastic properties, providing little evidence that viscosupplements can be evaluated in a pre-clinical context solely based on their rheological properties. The coefficient of friction of the **HA(270)-FA-TEGERA-CL** and **HA(270)-FA-HEGERA-CL** series’ frameworks was measured as a function of sliding distance (Figure 7).

Even though all samples showed very similar values, the **HA(270)-FA-TEGERA-CL** series exhibited lower coefficient of friction (COF) values compared to the **HA(270)-FA-HEGERA-CL** series. Despite the 3D crosslinked structure, both the **HA(270)-FA-TEGERA-CL** and **HA(270)-FA-HEGERA-CL** gel-like materials showed COF values similar to those of the main HA solutions used as viscosupplements (i.e., 0.25–0.15 at 10 mm/s) [96]. This aspect could be of significant importance, as the presence of a crosslinked structure may ensure a longer residence time, preventing hyaluronidase degradation, without affecting the coefficient of friction, which remains very close to that of commercial HA-based viscosupplements.

#### 2.3.5. In Vitro Degradation Test

The enzymatic resistance of crosslinked gel-like materials was measured and compared with two commercial viscosupplements. A significant increase in enzymatic resistance was found. Again, the polymer grafting degree played a major role in comparison with the crosslinking arm. Indeed, both **HA(270)-FA-TEGERA-CL-40** and **HA(270)-FA-HEGERA-CL-40** showed a higher resistance (Figure 8).

## 3. Conclusions

The CuAAC reaction was applied to graft copolymers (**HA-FA-Pg**) based on hyaluronic acid showing medium molecular weight (i.e., 270 kDa) grafted (at different extents) with ferulic acid moieties linking propargyl groups. In particular, the clickable alkyne groups of **HA(270)-FA-Pg** were exploited in a click-crosslinking reaction involving resorcinol-based cinnamate derivatives linking azido-terminated oligo(ethylene glycol) side chains. The CuAAC reaction was performed under very mild conditions, and the crosslinked gel-like materials, **HA(270)-FA-TEGERA-CL** and **HA(270)-FA-HEGERA-CL** series, were well characterized from the point of view of their structure and rheological features to evaluate their applicability as viscosupplements.

The modulable crosslinking density permitted reaching an adequate swelling capability for the foreseen application. In particular, **HA(270)-FA-TEGERA-CL** and **HA(270)-FA-HEGERA-CL** frameworks showed adequate swelling performance, the required shear thinning behavior, and coefficient of friction values very close to those of the main commercial HA solutions used as viscosupplements. Moreover, the presence of a crosslinked structure and remarkable resistance against hyaluronidase guaranteed a longer residence time. In particular, **HA(270)-FA-TEGERA-CL-40** and **HA(270)-FA-HEGERA-CL-40** appeared promising as viscosupplements since after 48 h they showed a four times greater enzymatic resistance than the commercial viscosupplements.

## 4. Materials and Methods

### 4.1. Synthesis and Characterization

*Trans*-3,5-dimethoxy cinnamic acid (**4**, ≥99%), phosphorus(V) oxychloride (POCl_3_, 99%), boron tribromide 1 M in dichloromethane (BBr_3_ 1 M in CH_2_Cl_2_), cesium carbonate (Cs_2_CO_3_, 99%), sodium iodide (NaI, ≥99%), tri(ethylene glycol) (**2a**, 99%), hexa(ethylene glycol) (**3a**, 97%), copper(II) sulfate pentahydrate (CuSO_4_·5 H_2_O ≥ 98%), sodium ascorbate (≥98%), methanesulfonyl chloride (CH_3_SO_2_Cl, ≥ 99.7%), sodium azide (NaN_3_, ≥99.5%), ethanol (EtOH), dichloromethane (CH_2_Cl_2_, ≥99.9%), petroleum ether (90%), ethyl acetate (≥99.5%), acetonitrile (CH_3_CN, 99.8%), *tert*-butanol (≥99%), acetone (≥99.5%), triethylethylenamine (TEA, ≥99.5%), and dimethylformammide (DMF, 99.8%) were purchased from Sigma-Aldrich (Taufkirchen, Germany) and were used as received without further purification.

Sodium salt of hyaluronic acid, with an average molecular weight (Mw) of 260 ± 10 kDa and a polydispersity index (Mw/Mn) of 3.1 ± 0.1 [HA(270)] was purchased from Biophil Italia SpA (Origgio, Italy), and used without further purifications [83].

Merck TLC aluminum sheets and silica gel 60 F_254_ were employed for TLC. NMR spectra of all the samples were recorded with either a Varian Mercury 300 (Varian, Palo Alto, CA, USA) or a Bruker DRX-600 AVANCE III (Bruker, Billerica, MA, USA) spectrometer in the indicated solvents. The chemical shift (*δ*) values are described in ppm and those of the H-H coupling constants (*J*) in Hz. Mass spectra were recorded with an Agilent 1100 LC/MSD employing an electrospray source.

Melting points were determined in open capillaries on a Gallenkamp (Beacon House, Nuffield Road, Cambridge, UK) apparatus and are uncorrected.

**HA(270)-FA-Pg** graft copolymers were synthesized by following the previously described procedure [90], starting from medium-molar-mass HA (1.0 g, 2.49 mmol in monomeric units).

2-(2-(2-Azidoethoxy)ethoxy)ethyl methanesulfonate (**3a**) and 17-azido-3,6,9,12,15-pentaoxaheptadecyl methanesulfonate (**3b**) were prepared by following the previously described three-step procedure [91] with the commercially available tri(ethylene glycol) **2a** or hexa(ethylene glycol) **2b**, respectively.

#### 4.1.1. Ethyl (E)-3-(3,5-Dimethoxyphenyl)acrylate (**5**)

The commercially available *trans*-3,5-dimethoxy cinnamic acid **4** (1.0 g, 4.80 mmol) was solubilized in dry EtOH (15 mL), and then phosphorus(V) oxychloride (0.67 mL, 7.20 mmol) was added dropwise. The resulting mixture was heated to reflux overnight in an inert nitrogen atmosphere. Then, the reaction mixture was concentrated under reduced pressure, and the organic residue was partitioned between dichloromethane and water. The combined organic extracts were dried over anhydrous sodium sulfate, filtered, and concentrated under reduced pressure. Purification of the crude residue by flash chromatography with petroleum ether–ethyl acetate (85:15) as the eluent gave compound **5** (1.08 g, yield 95%) as a white solid. Mp. 52.4–52.6 °C. ^1^H NMR (300 MHz, CD_3_OD) 1.35 (t, *J* = 7.1, 3H), 3.82 (s, 6H), 4.27 (q, *J* = 7.1, 2H), 6.51 (d, *J* = 16.0, 1H), 6.55 (t, *J* = 2.3, 1H), 6.77 (d, *J* = 2.3, 2H), 7.62 (d, *J* = 16.0, 1H). MS (ESI): *m*/*z* 259.1 [M + Na^+^] [97].

#### 4.1.2. Ethyl (E)-3-(3,5-Dihydroxyphenyl)acrylate (**6**)

To a cooled (0 °C) solution of previously obtained compound **5** (200 mg, 0.846 mmol) in dry CH_2_Cl_2_ (10 mL), a 1 M solution of boron tribromide in CH_2_Cl_2_ (5.1 mL, 5.1 mmol), was added and the reaction mixture was stirred at room temperature overnight under a nitrogen atmosphere. Then, the solvent was removed under reduced pressure, the crude residue was treated with a saturated solution of NaHCO_3_, and the organic layer was extracted with ethyl acetate. The combined extracts were then dried over anhydrous sodium sulfate, filtered, and the solvent concentrated under reduced pressure. The crude product was purified by means of flash chromatography using petroleum ether–ethyl acetate (8:2) as the eluent to yield compound **6** (138 mg, yield 78%) as a white solid. Mp. 136.3–136.5 °C. ^1^H NMR (300 MHz, CD_3_OD): 1.31 (t, *J* = 7.1, 3H), 4.22 (q, *J* = 7.1, 2H), 6.31 (t, *J* = 2.2, 1H), 6.35 (d, *J* = 16.0, 1H), 6.49 (d, *J* = 2.1, 2H), 7.49 (d, *J* = 16.0, 1H). MS (ESI): *m*/*z* 231.1 [M + Na^+^].

#### 4.1.3. Ethyl (E)-3-(3,5-Bis(2-(2-(2-Azidoethoxy)ethoxy)ethoxy)phenyl)acrylate (**1a**)

Derivative **6** (73 mg, 0.35 mmol) was solubilized in dry CH_3_CN (16 mL), and then Cs_2_CO_3_ (459 mg, 1.41 mmol), NaI (104 mg, 0.69 mmol), and derivative **3a** (58 mg, 0.23 mmol) were added. The reaction mixture was heated to reflux in an inert nitrogen atmosphere for 24 h. Then, the solvent was removed in vacuo to obtain an organic crude, which was solubilized in CH_2_Cl_2_ and treated with 1N HCl solution. Then, anhydrous Na_2_SO_4_ was added, and the organic layer was filtered and concentrated under reduced pressure. The residue was purified by flash chromatography with petroleum ether–ethyl acetate (6:4) as eluent to yield compound **1a** (33 mg, yield 55%) as a yellow oil. ^1^H NMR (600 MHz, CD_3_OD): 1.34 (t, *J* = 7.1, 3H), 3.38 (t, *J* = 4.9, 4H), 3.66–3.75 (m, 12H), 3.88 (t, *J* = 4.6, 4H), 4.17 (t, *J* = 4.6, 4H), 4.26 (q, *J* = 7.1, 2H), 6.52 (d, *J* = 15.9, 1H), 6.63 (t, *J* = 2.3, 1H), 6.82 (d, *J* = 2.3, 2H), 7.62 (d, *J* = 15.9, 1H). MS (ESI): *m*/*z* 545.2 [M + Na^+^].

#### 4.1.4. Ethyl (E)-3-(3,5-Bis((17-Azido-3,6,9,12,15-pentaoxaheptadecyl)oxy)phenyl)acrylate (**1b**)

Derivative **6** (102 mg, 0.49 mmol) was solubilized in dry CH_3_CN (15 mL), and then Cs_2_CO_3_ (635 mg, 1.95 mmol), NaI (145 mg, 0.97 mmol), and derivative **3b** (125 mg, 0.32 mmol) were added. The reaction mixture was heated to reflux in an inert nitrogen atmosphere for 24 h. Then, the solvent was removed in vacuo to obtain an organic crude, which was solubilized in CH_2_Cl_2_ and treated with 1N HCl solution. Then, anhydrous Na_2_SO_4_ was added, and the organic layer was filtered and concentrated under reduced pressure. The residue was purified by flash chromatography with ethyl acetate–methanol (95:5) as the eluent to furnish compound **1b** (80 mg, yield 63%) as a yellow oil. ^1^H NMR (600 MHz, CD_3_OD): 1.35 (t, *J* = 7.1, 3H), 3.38 (t, *J* = 4.9, 4H), 3.59–3.78 (m, 36H), 3.87 (t, *J* = 4.6, 4H), 4.17 (t, *J* = 4.6, 4H), 4.27 (q, *J* = 7.1, 2H), 6.53 (d, *J* = 16.0, 1H), 6.63 (t, *J* = 2.3, 1H), 6.82 (d, *J* = 2.3, 2H), 7.62 (d, *J* = 16.0, 1H). MS (ESI): *m*/*z* 809.3 [M + Na^+^].

### 4.2. Click Chemistry Crosslinking General Procedure Involving HA(270)-FA-Pg Graft Copolymers

A 10 mL flask was charged (under an inert atmosphere) with *tert*-butanol (2.0 mL), water (2.0 mL), and a solution of CuSO_4_ pentahydrate (12.5 mg, 0.050 mmol) in 0.50 mL of water. A 1 M solution of sodium ascorbate in water (0.50 mL) was then added and 1.0 mL of the resulting mixture was used as the catalyst. A mixture of the appropriate medium-molecular-weight **HA(270)-FA-Pg** graft copolymer (250 mg) and the crosslinking agent **1a** (see the amounts in Table 2) or 1**b** (see the amounts in Table 3) [90] in *tert*-butanol (25 mL) and water (25 mL) was treated with the catalyst solution (1.0 mL) and the resulting reaction mixture was stirred overnight at room temperature. The solvent was then removed in vacuo. Purification of the gel residue by washing with acetone (5 × 50 mL) gave the expected crosslinked materials, which were dried under reduced pressure.

HA(270)-FA-TEGERA-CL-10, HA(270)-FA-TEGERA-CL-20, and HA(270)-FA-TEGERA-CL-40 Materials

**Table 2 gels-10-00751-t002:** Reaction parameters in the crosslinking of **HA(270)-FA-Pg** graft copolymers with crosslinking agent **1a** to give **HA(270)-FA-TEGERA** gel-like materials.

Copolymer	mg	1a (mg)	1a (mmol)	Gel-like Materials	mg Obtained	Appearance of Obtained Solid
**HA(270)-FA-Pg-10**	250	19.0	0.037	**HA(270)-FA-TEGERA-CL-10**	260 ^a^	pale-yellow
**HA(270)-FA-Pg-20**	250	36.0	0.069	**HA(270)-FA-TEGERA-CL-20**	283 ^b^	off-white
**HA(270)-FA-Pg-40**	250	64.0	0.123	**HA(270)-FA-TEGERA-CL-40**	314	off-white

^a 1^H NMR (600 MHz, D_2_O) spectrum is depicted in Appendix A. ^b 1^H NMR (600 MHz, D_2_O) spectrum is depicted in Figure 2.

HA(270)-FA-HEGERA-CL-10, HA(270)-FA-HEGERA-CL-20, and HA(270)-FA-HEGERA-CL-40 Materials

**Table 3 gels-10-00751-t003:** Reaction parameters in the crosslinking of **HA(270)-FA-Pg** graft copolymers with crosslinking agent **1b** to give **HA(270)-FA-HEGERA** gel-like materials.

Copolymer	mg	1b (mg)	1b (mmol)	Gel-like Materials	mg Obtained	Appearance of Obtained Solid
**HA(270)-FA-Pg-10**	250	24.0	0.031	**HA(270)-FA-HEGERA-CL-10**	246 ^a^	off-white
**HA(270)-FA-Pg-20**	250	54.0	0.069	**HA(270)-FA-HEGERA-CL-20**	290 ^b^	off-white
**HA(270)-FA-Pg-40**	250	97.0	0.123	**HA(270)-FA-HEGERA-CL-40**	316	off-white

^a 1^H NMR (600 MHz, D_2_O) spectrum is depicted in Appendix A. ^b 1^H NMR (600 MHz, D_2_O) spectrum is depicted in Figure 3.

### 4.3. Swelling Behavior

The swelling behavior of all the frameworks was analyzed in terms of swelling kinetics and water types, i.e., freezing (_f_W) and non- freezing water (_nf_W). Three replicates were used for each analysis. The swelling kinetics was measured by dipping 20 mg of the dry state samples in water at 37 °C. The weight increase was measured for 8 h (every 30 min), and then at 24 h, 48 h, and 72 h, with the swelling ratio as defined in Equation (1) [98]:SR% = [(Ws − Wd)/Wd] × 100(1)

In detail, Wd and Ws represent, respectively, the weight of the dry and swollen framework.

### 4.4. Thermal Analysis

Thermogravimetric analysis (TGA) was performed using a SDT-Q600 (TA Instruments, Leatherhead, UK) and differential scanning calorimetry (DSC) tests were conducted with a DSC Q1000 (TA Instruments, Leatherhead, UK); these techniques were used to quantify the water types, i.e., freezing (_f_W) and non-freezing water (_nf_W) of the gel-like materials.

The total water content (W_TOT_), or the sum of freezing and non-freezing water of each sample, was assessed using TGA. In detail, 10 mg of full swollen in ultra-pure water (UPW) frameworks were heated at a heating rate of 10 °C/min, starting from 30 °C to the final temperature of 300 °C using an inert atmosphere (N_2_, 100 mL/min). The total water content is expressed in Equation (2).
W_TOT_ = _f_W + _nf_W(2)
where W_TOT_, _f_W, and _nf_W are, respectively, the total, the freezable, and the non-freezable water weights contained in the samples.

Freezing water was calculated by DSC sealing about 5 mg of each UPW sample in an aluminum hermetic anodized pan. The samples were analyzed using a heating rate of 0.2 °C/min and the cell was kept under nitrogen (50 mL/min). The method steps consist of a first colling one, from 40 °C to −40 °C, followed by an isotherm for 5 min. Finally, the samples were heated up from −40 °C to a final temperature of +40 °C. All the steps of the method were performed under N_2_ flow at 50 mL/min. The weight of the freezable water is obtained by the enthalpy of melting of the frozen water contained in the sample (ΔHm) and by the latent heat of melting of pure water (ΔH). The ΔHm/ΔH ratio is associated with the weight of freezable water per weight of the corresponding full swollen sample (W_SG_) [99]. W_SG_ was measured for all samples once swelling equilibrium was reached, i.e., after 48 h of immersion. Applying Equation (3) makes it possible to calculate the _f_W:_f_W/W_SG_ = ∆Hm/∆H(3)

_nf_W can be obtained by the difference using Equation (2).

Freezable water (_f_W) can be distinguished in free freezable water (_ff_W) and bound freezable water (_bf_W), using the heating rate (HR) signal in the resulting thermograms [100].

Mesoporosity: 10–20 mg of each gel-like material in a swollen state was sealed in aluminum pans that were cooled to −60 °C, then the sample was heated up to a temperature of −0.3 °C and kept for 10 min isothermally. After that, the material was cooled till −60 °C with a HR of 0.2 °C/min [101].

Using Equation (4), the pore radius can be obtained:Rp nm = (−64.67/ΔT) + 0.57(4)
where ΔT is the difference with the triple-point temperature.

### 4.5. Thermal Stability

Thermal stability was measured by TGA on 10 mg of dry state sample, heating it from the temperature of 30 °C to 850 °C, at a HR of 10 °C/min in under N_2_ (100 mL/min).

### 4.6. Viscoelastic Properties

The mechanical behavior of crosslinked gel-like materials was evaluated at 37 °C in the swollen state (NaCl 154 mM). The analyses were on a Discovery Hybrid Rheometer-2 (DHR-2) (TA Instruments, Leatherhead, UK) equipped with a Peltier temperature control system, using a plate–plate stainless steel geometry. On the gel-like materials were performed a preliminary strain sweep test, to evaluate the Linear Viscoelastic Region (LVR). The test was performed in the strain range from 0.01% to 5%, at the constant frequencies of 0.01 Hz, 1 Hz and 15 Hz. The mechanical properties of the materials were evaluated from a frequency sweep test in shear and on a compression regime: those tests were performed at a constant strain of 0.05% (within the LVR, as deduced from the strain sweep test), in the range of frequency from 0.01 Hz to 15 Hz. Viscosity of samples were evaluated by flow sweep experiments: the test was performed with a ramp from 0.01 s^−1^ to 250 s^−1^ of the shear rate [102].

### 4.7. Tribological Properties

The tribological behavior of the materials was evaluated at 37 °C in the swollen state (NaCl 154 mM). The analyses were on a Discovery Hybrid Rheometer-2 (DHR-2) (TA Instruments, Leatherhead, UK) using a three-balls-on-plate geometry (three 5/16 stainless steel truncated balls). From this test, the Coefficient Of Friction (COF) curve was evaluated at a loading force of 4.5 N; to achieve this, a flow sweep test in the range of shear rate from 0.001 rad/s to 10 rad/s was performed.

### 4.8. In Vitro Degradation Test

Enzymatic degradation by hyaluronidase was evaluated using the procedure reported by Saad et al. [103]. Briefly, 100 mg/L solutions of each sample in ammonium acetate 10% *w*/*w* were prepared and maintained at 37 °C. A 125 μL of 5.8 mU solution of hyaluronidase was added to 1 mL of each solution and kept at 37 °C for 48 h, taking aliquots after 1 h, 3 h, 24 h, and 48 h. Each aliquot was analyzed by spectrophotometric analysis measuring with a UV/Visible spectrophotometer (Lambda 25, Perkin Elmer, Waltham, MA, USA) the absorbance at 530 nm to quantify the degradation to uronic acid of hyaluronic acid as a function of time [104].

## Data Availability

Data are available upon request to the corresponding author(s).

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
