# Peer review of "Crosslinking by Click Chemistry of Hyaluronan Graft Copolymers Involving Resorcinol-Based Cinnamate Derivatives Leading to Gel-like Materials"

_gels, 2024, doi:10.3390/gels10110751_

Round 1
Reviewer 1 Report
Comments and Suggestions for Authors
The work entitled " Cross-Linking by Click-Chemistry of Hyaluronan Graft 2 Copolymers Involving Resorcinol-Based Cinnamate 3 Derivatives Leading to Gel-like Materials" is interesting and worthy of publication. However, there are some issues, which need to be resolved before I can recommend its publication. The following points need to be addressed:
1. Abstract
- The abstract is poorly written. There are complex and long sentences need to be simplified. The objective of the study must be clarified.
- The abstract must include How did authors characterize the synthesized crosslinked structures?
2. Introduction
- Provide the open name of ECM.
- Quality of fig. S1 is poor and not readable. Also, Fig. 1 should be improved.
- The novelty and the research gap are not clear. Could you please make the introduction a bit more comprehensive? What the advantages of crosslinking by click chemistry over the other methods?
3. Experimental Section
- What are the reagents and solvents used? Better to mention them in the text.
- Provide a section for the spectral characterization. The authors performed only HNMR and mass spectroscopy for characterization of the synthesized compounds. Why was not FTIR used?
- What is the melting point of compounds 5&6, compared to the starting acid 4.
- Authors should clarify how the HA(270)-FA-Pg graft copolymers was synthesized?
- Swelling Behavior: What do water types stand for? The sentence “The swelling kinetics was measured dipping about 20 mg….” correct to “…..measured by dipping…..”. For accuracy and reliability, the swelling experiments should be performed at least three times, and the average is taken.
- Equation 1 is wrong. You can take help from [International Journal of Biological Macromolecules 159 (2020) 422–432]
- Lines 221, 227, 330, check the structure of sentences.
4. Results and Discussion
- Figure 1. and Scheme 2. are approximately similar.
- The crosslinked gels do not soluble in any solvents. So, I am a bit worried about the recorded HNMR in D2O. Why was it not performed using solid NMR?
- Swelling ratio of hydrogels are usually strongly influenced by degree of crosslinking. This must be discussed clearly in Section 3.3.1 for HA(270)-FA-TEGERA-CL-10,20,40 and HA(270)-FA-HEGERA-CL-10,20,40.
- Figure of the swelling ratio Vs. time must be presented.
- There are many complicated sentences which must be simplified, i.e. lines 371-376, 400-405, etc
- Results of the rheological analysis need to be discussed in more details supported with the relevant references.
- Head of section 3.3.5. italicize “In Vitro”.
5. Conclusion
- The conclusion could be strengthened by summarizing the main numerical findings.
6. References
- Only 12 refs out 95 in the last three years, so updating the references is appreciated.
7. Again, authors should thoroughly check the manuscript for many grammatically errors.
Comments on the Quality of English LanguageThe English must be improved to more clearly express the research.
Author Response
To: Ms. Brynn Zhou
Assistant Editor
Gels Editorial Office
E-Mail: Brynn.zhou@mdpi.com
Manuscript ID: gels-3272816
Dear Ms. Zhou,
On behalf of the authors, I submit a revised version of the manuscript "Cross-Linking by Click-Chemistry of Hyaluronan Graft Copolymers Involving Resorcinol-Based Cinnamate Derivatives Leading to Gel-like Materials" by Mario Saletti, Simone Pepi, Marco Paolino, Jacopo Venditti, Germano Giuliani, Claudia Bonechi, Gemma Leone, Agnese Magnani, Claudio Rossi, and Andrea Cappelli.
The original manuscript has been adapted by following the standard of Gels, and it has been revised after carefully considering the comments of three independent reviewers who recommended publication after revision.
A detailed reply to the reviewers is attached below, and the changes are highlighted in blue text in the submitted manuscript.
I now hope that you will be able to accept our manuscript for publication in Gels.
Best Regards,
Andrea Cappelli
Response to Reviewer 1
The work entitled " Cross-Linking by Click-Chemistry of Hyaluronan Graft 2 Copolymers Involving Resorcinol-Based Cinnamate 3 Derivatives Leading to Gel-like Materials" is interesting and worthy of publication. However, there are some issues, which need to be resolved before I can recommend its publication. The following points need to be addressed:
- Abstract
- The abstract is poorly written. There are complex and long sentences need to be simplified. The objective of the study must be clarified.
Authors’ answer: The abstract has been simplified and the objective of this study has been emphasized according to the reviewer’s suggestion.
- The abstract must include How did authors characterize the synthesized crosslinked structures?
Authors’ answer: We thank the reviewer for the suggestion. The methods employed for the characterization of newly-synthesized crosslinked structures have been explained.
- Introduction
- Provide the open name of ECM.
Authors’ answer: We thank the reviewer for the suggestion. The open name of ECM has been added in the manuscript.
- Quality of fig. S1 is poor and not readable. Also, Fig. 1 should be improved.
Authors’ answer: We thank the reviewer for the suggestions. Figure S1 and Figure 1 have been improved.
- The novelty and the research gap are not clear. Could you please make the introduction a bit more comprehensive? What the advantages of crosslinking by click chemistry over the other methods?
Authors’ answer: We thank the reviewer for the useful suggestions. The introduction of this manuscript has been improved, especially in the novelty and the advantages of crosslinking by click chemistry methods. The innovation is in the structure of the crosslinking agents used in the CuAAC reaction.
- Experimental Section
- What are the reagents and solvents used? Better to mention them in the text.
Authors’ answer: We thank the reviewer for the useful suggestions. Reagents and solvents have been mentioned in paragraph 4 Materials and Methods.
- Provide a section for the spectral characterization. The authors performed only HNMR and mass spectroscopy for characterization of the synthesized compounds. Why was not FTIR used?
Authors’ answer: NMR analysis enables the evaluation of reorientation dynamics in macromolecular systems in solution, which is critical for studying the synthesis of polymers and highlighting the supramolecular interactions. In this study, the formation of crosslinked structures was also confirmed by the significant broadening of proton signals, indicative of the system's low mobility within the lattice structure. Although FT-IR spectroscopy is essential for identifying functional chemical groups, it was not suitable for assessing the formation of crosslinked materials showing high molecular weight. The disappearance of proton signals at low fields also enables the monitoring of the formation of the crosslinked material through a straightforward comparison of the spectra.
- What is the melting point of compounds 5&6, compared to the starting acid 4.
Authors’ answer: Compounds 5 and 6 are known in the literature and were synthesized as reported in the paper. The appropriate references and the relevant melting points have been added in the revised manuscript.
- Authors should clarify how the HA(270)-FA-Pg graft copolymers was synthesized?
Authors’ answer: We thank the reviewer for the suggestion. HA(270)-FA-Pg graft copolymers were synthesized employing the previously reported procedure (see ref. 90).
- Swelling Behavior: What do water types stand for?
Authors’ answer: Water types have been explicated: “The swelling behavior of all the frameworks was analyzed in terms of swelling kinetics and water types, i. e. freezing (fW) and non- freezing water (nfW).”
- The sentence “The swelling kinetics was measured dipping about 20 mg….” correct to “…..measured by dipping…..”.
Authors’ answer: The sentence “The swelling kinetics was measured dipping about 20 mg….” has been corrected to “The swelling kinetics was measured by dipping about 20 mg of each sample in dry state in an excess of water at 37 °C”.
- For accuracy and reliability, the swelling experiments should be performed at least three times, and the average is taken
Authors’ answer: Authors apologize for this. All the reported numerical results are the average of three replicates. The sentence “ 3 replicates were used for each analysis” has been added.
- Equation 1 is wrong. You can take help from [International Journal of Biological Macromolecules 159 (2020) 422–432]
Authors’ answer: Authors apologize for this. It was a typo. Manuscript has been corrected.
- Lines 221, 227, 330, check the structure of sentences.
Authors’ answer: We thank the reviewer for the useful suggestion. The sentences have been improved.
- Results and Discussion
- Figure 1. and Scheme 2. are approximately similar.
Authors’ answer: Figure 1 has been modified to be dissimilar to Scheme 2 according to the useful reviewer’s suggestion.
- The crosslinked gels do not soluble in any solvents. So, I am a bit worried about the recorded HNMR in D2O. Why was it not performed using solid NMR?
Authors’ answer: We thank the reviewer for the suggestion. However, despite lower local mobility in the HA chains of the crosslinked derivatives, the resulting gel-like materials were quite soluble in water, which was selected as a biocompatible solvent, so we recorded the 1H NMR in D2O.
- Swelling ratio of hydrogels are usually strongly influenced by degree of crosslinking. This must be discussed clearly in Section 3.3.1 for HA(270)-FA-TEGERA-CL-10,20,40 and HA(270)-FA-HEGERA-CL-10,20,40.
Authors’ answer: Section 3.3.1 has been modified.
- Figure of the swelling ratio Vs. time must be presented.
Authors’ answer: In SI, a new figure (i. e. Figure S4) has been added which shows the swelling kinetics of prepared hydrogels.
- There are many complicated sentences which must be simplified, i.e. lines 371-376, 400-405, etc.
Authors’ answer: We thank the reviewer for the useful suggestions. The sentences have been revised and simplified.
- Results of the rheological analysis need to be discussed in more details supported with the relevant references.
Authors’ answer: Rheological analysis has been modified and relevant references have been added.
- Head of section 3.3.5. italicize “In Vitro”.
Authors’ answer: The head of section 3.3.5 has been changed in section 4.8 and it has been italicized in agreement with the reviewer’s suggestion.
- Conclusion
- The conclusion could be strengthened by summarizing the main numerical findings.
Authors’ answer: We thank the reviewer for the useful suggestion. The conclusion has been improved.
- References
- Only 12 refs out 95 in the last three years, so updating the references is appreciated.
Authors’ answer: We thank the reviewer for the suggestion. Some last three years' references have been added.
- Again, authors should thoroughly check the manuscript for many grammatically errors.
Authors’ answer: We thank the reviewer for the appreciate suggestion. We thoroughly checked the manuscript again, and the English has been improved.
Please see the attachment.

Reviewer 2 Report
Comments and Suggestions for Authors
In this manuscript by Cappelli et al., the authors reported the synthesis of crosslinked materials by using a Click-Chemistry approach. The synthesized materials were characterized from the point of view of their structure and rheological features in order to evaluate their potential gel-like properties. The manuscript is well written and results are also interesting. Experiments have been well designs and conclusions reflect the findings reported in the experimental section. The manuscript is ready to be accepted in present form.
Author Response
Response to Reviewer 2
In this manuscript by Cappelli et al., the authors reported the synthesis of crosslinked materials by using a Click-Chemistry approach. The synthesized materials were characterized from the point of view of their structure and rheological features in order to evaluate their potential gel-like properties. The manuscript is well written and results are also interesting. Experiments have been well designs and conclusions reflect the findings reported in the experimental section. The manuscript is ready to be accepted in present form.
Authors’ answer: We thank the reviewer for the general appreciation of our work.
Please see the attachment.

Reviewer 3 Report
Comments and Suggestions for Authors
Dear Authors,
You have prepared a very dense paper that is difficult to read. After several readings, it keeps still unclear to me where the innovation is. You have tried to prepare several first adducts and after crosslinking using an specific catalyst based on Cu.
In my opinion, this format with bold everywhere causes distortion when reading.
Reagents and instruments should clearly specify city and Country as you did only in the case of TGA (on page 6). Please, consider adding all this information.
The data in lines 121,122 should be given after the experimental spectra. First, every spectra should be added and after, describe the interesting values to consider. Please, take that into account when revising the article
On page 4, lines 133 to 136 and 145 to 136, as well as lines 158 to 162. Are affected by the same mistake.
When describing the different procedures to get the previous polymers, please specify at the end the complete name of the compound obtained. This format using 1,2, 3, and a,b, and c is very confusing for readers.
It should be, probably easier for readers to follow the procedures if you include after them, the reaction squeme that you've used in results.
Page 5. To me gives unuseful information. Please, consider to eliminate, or to join in an unique paragraph.
It would be interesting to see the kinetics of swelling with times. Wsg is obtained, at which time?? The same for all samples?? Please, clarify
page 6. Please, consider to give more space among paragraphs when you introduce inside one formulae. Very difficult to follow the equations .
Fig 4 to Fig 8. Please, consider to reduce the size of the legends. Aesthetically bad effect.
Reference 41, please consider to uniform with the rest.
MAny thanks
Comments on the Quality of English LanguageThe english language used in the article is useful enough to follow the descriptions of methodologies and characterizations
Author Response
To: Ms. Brynn Zhou
Assistant Editor
Gels Editorial Office
E-Mail: Brynn.zhou@mdpi.com
Manuscript ID: gels-3272816
Dear Ms. Zhou,
On behalf of the authors, I submit a revised version of the manuscript "Cross-Linking by Click-Chemistry of Hyaluronan Graft Copolymers Involving Resorcinol-Based Cinnamate Derivatives Leading to Gel-like Materials" by Mario Saletti, Simone Pepi, Marco Paolino, Jacopo Venditti, Germano Giuliani, Claudia Bonechi, Gemma Leone, Agnese Magnani, Claudio Rossi, and Andrea Cappelli.
The original manuscript has been adapted by following the standard of Gels, and it has been revised after carefully considering the comments of three independent reviewers who recommended publication after revision.
A detailed reply to the reviewers is attached below, and the changes are highlighted in blue text in the submitted manuscript.
I now hope that you will be able to accept our manuscript for publication in Gels.
Best Regards,
Andrea Cappelli
Response to Reviewer 3
Dear Authors,
You have prepared a very dense paper that is difficult to read. After several readings, it keeps still unclear to me where the innovation is. You have tried to prepare several first adducts and after crosslinking using an specific catalyst based on Cu.
Authors’ answer: The innovation is in the structure of the crosslinking agents used in the CuAAC reaction.
In my opinion, this format with bold everywhere causes distortion when reading.
Authors’ answer: A lot of bold was deleted to simplify the reading of the manuscript according to the reviewer’s suggestion. Now, only the compounds and materials involved in the present paper are in bold format.
Reagents and instruments should clearly specify city and Country as you did only in the case of TGA (on page 6). Please, consider adding all this information.
Authors’ answer: City and Country of reagents and instruments have been added according to the reviewer’s suggestion.
The data in lines 121,122 should be given after the experimental spectra. First, every spectra should be added and after, describe the interesting values to consider. Please, take that into account when revising the article
Authors’ answer: The data in lines 121 and 122 represented the complete list of the peaks appearing in the 1H NMR spectrum of compound 5 and were used in the main text in substitution of the full 1H NMR spectrum.
On page 4, lines 133 to 136 and 145 to 136, as well as lines 158 to 162. Are affected by the same mistake.
Authors’ answer: The data in lines 133 to 136 represented the complete list of the peaks appearing in the 1H NMR spectrum of compound 6 and were used in the main text in substitution of the full 1H NMR spectrum. The data in lines 145 to 148 represented the complete list of the peaks appearing in the 1H NMR spectrum of compound 1a and were used in the main text in substitution of the full 1H NMR spectrum. The data in lines 158 to 162 represented the complete list of the peaks appearing in the 1H NMR spectrum of compound 1b and were used in the main text in substitution of the full 1H NMR spectrum.
When describing the different procedures to get the previous polymers, please specify at the end the complete name of the compound obtained. This format using 1,2, 3, and a,b, and c is very confusing for readers.
It should be, probably easier for readers to follow the procedures if you include after them, the reaction squeme that you've used in results.
Authors’ answer: We were unable to understand this suggestion.
Page 5. To me gives unuseful information. Please, consider to eliminate, or to join in an unique paragraph.
Authors’ answer: We thank the reviewer for the useful suggestion. To simplify the information about the synthetic procedures and the obtained results Table 2 and Table 3 have been added.
It would be interesting to see the kinetics of swelling with times. Wsg is obtained, at which time?? The same for all samples?? Please, clarify
Authors’ answer: Wsg was measured for all samples once swelling equilibrium was reached, i.e., after 48 hours of immersion. This information has been added to the manuscript.
A new figure (Figure S4a-b) which shows the swelling kinetics of prepared hydrogels has been added in SI.
page 6. Please, consider to give more space among paragraphs when you introduce inside one formulae. Very difficult to follow the equations.
Authors’ answer: We thank the reviewer for the useful suggestion. The spaces between paragraphs and the formulae have been added to simplify the reading of the manuscript.
Fig 4 to Fig 8. Please, consider to reduce the size of the legends. Aesthetically bad effect.
Authors’ answer: Figures 4 to 8 have been changed accordingly with reviewer suggestion
Reference 41, please consider to uniform with the rest.
Authors’ answer: Reference 41 has been revised in agreement with the reviewer’s suggestion.
Please see the attachment.

Round 2
Reviewer 1 Report
Comments and Suggestions for Authors
The quality of the manuscript has been improved during the revision process. And now it is suitable for the publication by Gels.
Reviewer 3 Report
Comments and Suggestions for Authors
Dear Authors,
Now, the article looks much better after revision process. You have made the required changes by reviewers, and you have corrected several mistakes in the text.
Many thanks